# The Effect of Serum Leptin Concentration and Leptin Receptor Expression on Colorectal Cancer

**DOI:** 10.3390/ijerph20064951

**Published:** 2023-03-11

**Authors:** Sylwia Chludzińska-Kasperuk, Jolanta Lewko, Regina Sierżantowicz, Elżbieta Krajewska-Kułak, Joanna Reszeć-Giełażyn

**Affiliations:** 1Biobank, Medical University of Bialystok, 15-269 Bialystok, Poland; 2Department of Primary Health Care, Medical University of Bialystok, 15-054 Bialystok, Poland; 3Department of Surgical Nursing, Medical University of Bialystok, 15-274 Bialystok, Poland; 4Department of Integrated Medical Care, Medical University of Bialystok, 15-089 Bialystok, Poland; 5Department of Medical Pathomorphology, Medical University of Bialystok, 15-269 Bialystok, Poland

**Keywords:** obesity, leptin, leptin receptor expression, colorectal cancer

## Abstract

Introduction: The level of leptin in the blood shows a positive, strong correlation with the mass of adipose tissue. Being overweight and having metabolic disorders increase the risk of developing colorectal cancer. Aim of the Paper: The aim of the study was to assess the concentration of leptin in the blood serum as well as the expression of the leptin receptor in colorectal cancer cells. In addition, the effect of serum leptin concentration and leptin receptor expression on clinical and pathological parameters such as BMI, obesity, TNM, and tumor size was assessed. Methods: The study included 61 patients diagnosed with colorectal cancer and treated with surgery. Results: Strong leptin receptor expression and the prevalence of overweight and obesity are factors influencing the occurrence of excessive leptin concentrations. Conclusion: Leptin may be involved in the development and progression of colorectal cancer. More research is needed to better elucidate the role of leptin in the development and progression of the disease.

## 1. Introduction

Colorectal cancer (CRC) is the third most common cancer and the fourth most common cause of death in the world. In addition to the aging of the population and eating habits, adverse risk factors, such as obesity, a lack of physical activity, and smoking, increase the risk of CRC [1,2,3]. Advances in the pathophysiology of CRC have increased the number of treatment options for CRC and led to the creation of individual treatment plans for patients [1,3,4]. Although the new treatment options have doubled the overall survival for advanced disease to 3 years, survival for CRC patients is still the best for those with non-metastatic disease. Since the disease only shows symptoms at an advanced stage, worldwide screening programs are being implemented to increase early detection and reduce the incidence and mortality rate of CRC [1,3,4,5]. The early diagnosis of CRC is very important, as it allows the cancer to be detected at an early stage, when there is a very good chance of a cure. Colonoscopy is the gold standard in the diagnosis of CRC. It has high diagnostic accuracy and can determine the exact location of the tumor. Importantly, this technique allows the simultaneous collection of tumor tissue material and, thus, the histological confirmation of the diagnosis and the collection of material for molecular tests [6]. New treatment options for colorectal cancer include immunohistochemical and histoenzymatic techniques (immunohistochemical identification of cytosolic cytochrome, histoenzymatic analysis of the activities of energy metabolism enzymes) [7]. In the diagnosis of CRC, radiology has various applications, such as ultrasound, computed tomography, positron emission tomography, and magnetic resonance imaging. Depending on the stage of CRC, a combination of treatments can be used: surgery, chemoimmunotherapy, and radiotherapy. Therefore, it is advisable not only to coordinate multidisciplinary treatment but also to quickly implement diagnostic procedures [8,9].

It is the second most common cancer diagnosed in women and the third among men. In women, the incidence and mortality of CRC are approximately 25% lower than in men. These indicators vary geographically, with the highest rates observed in the most developed countries. Stabilizing tendencies and decreasing incidence of CRC are observed only in highly developed countries [3,9]. Environmental factors are associated with the occurrence of cancer and account for approximately 83% of all colorectal cancer cases. Among these patients, the only environmental impact is detected in 30%. In 28% of them, the cancer is associated with epigenetic mutations, such as DNA repair genes; the remaining 25% of patients have a positive family history of CRC [10,11,12]. Being overweight and having metabolic disorders increase the risk of developing CRC. Studies indicate a clear correlation between body mass index (BMI) and the incidence of CRC. The risk of CRC increases with an increase in BMI in the range of 23 to 30 kg/m^2^. In adults with a BMI above 30 kg/m^2^ compared with people with a BMI of less than 23 kg/m^2^, this risk increases by 50–100%. An increase in BMI of 5 kg/m^2^ significantly increases the risk of CRC, and in people with a BMI above 30 kg/m^2^, it is already approximately 40% [5,13,14,15]. The exact mechanism by which obesity increases the risk of developing CRC is not known. Scientific research takes into account the mitogenic effects of insulin, insulin resistance, and hyperinsulinemia. The role of insulin in the carcinogenesis of CRC may result from its direct or indirect effect on increasing the concentration of IGF-1 in the blood serum, which stimulates proliferation by inhibiting the apoptosis of mutant cells [15,16,17].

Many epidemiological data prove that obesity increases the risk of CRC. Although the molecular mechanisms underlying this compound are not known, data from in vitro experiments suggest a direct contribution of adipose tissue to the development of CRC.

The main factor affecting serum leptin levels is body fat mass. The level of leptin in the blood shows a positive, strong correlation with body fat mass. In people with obesity, the level of leptin in the blood is almost 10 times higher than in people with a normal body weight. The results of research in recent years indicate that leptin is also produced by cancer cells of the large intestine and breast [18,19,20,21,22]. Numerous epidemiological studies indicate obesity as a significant risk factor conducive to the initiation of the cancer process as well as a prognostic factor associated with prognosis in people affected by cancer. Obesity has been proven to be an important prognostic factor for cancers of the mammary gland, uterine body, colon, and prostate. Given the positive correlation of leptin levels with body fat mass, more research should be conducted to investigate the contribution of leptin to the tumorigenesis process [18,19,20,21,22].

The aim of the study was to assess the concentration of leptin in the blood serum as well as the expression of the leptin receptor in colorectal cancer cells. In addition, the effect of serum leptin concentration and leptin receptor expression on clinical and pathological parameters such as BMI, obesity, TNM and tumor size was assessed. The study attempted to clarify the role of leptin as a new biomarker for CRC.

## 2. Material and Methods

The study involved a total of 61 patients successively admitted to the II Clinic of General and Gastroenterological Surgery of the University Clinical Hospital in Bialystok with a diagnosis of CRC treated surgically in the period 2018–2020.

The study was conducted after obtaining written consent from patients regarding biological material (blood and tumor tissue) and the clinical data of patients with CRC. Patients diagnosed with CRC were included in the study, whereas patients with a previous history of cancer and patients using hormonal treatments were excluded.

### 2.1. Experimental Designs

For the research, 1.6 mL of serum was stored at −80 °C till the tests were performed, and peripheral cancer tissue stored in paraffin blocks was used. The comparison group for serum leptin values consisted of a total of 60 patients (33 females and 27 males aged 35–60 years): 30 patients without cancer with a normal BMI who were not using hormonal treatment, and 30 patients without cancer with obesity I° BMI > 35, who were also not using hormonal therapy. The control to assess the expression of the leptin receptor in the tissue consisted of slices of normal intestinal mucosa taken from 20 patients (8 females and 12 males aged 45–65 years with a normal BMI) with diverticulosis of the large intestine stored in paraffin blocks. Clinical–pathological data were also analyzed. Serum leptin concentrations were tested using the Human Leptin ELISA Kit.

Immunohistochemical methods were used to evaluate leptin expression in peripheral tumor tissues. After resection of the large intestine, the collected tissue samples were fixed in a 10% buffered formaldehyde solution, then immersed in paraffin blocks and stained with hematoxylin and eosin. Immunohistochemical studies were performed to evaluate the expression of the leptin receptor in colorectal cancer samples and normal colorectal mucosa samples.

The tissue thickness was 4 μm. Leptin was investigated in representative tissue sections using specific antibodies for leptin and polyclonal antibodies A-20. All primary antibodies were diluted in phosphate-buffered saline with 1.5% normal blocking serum. The antibody–antigen reaction was revealed using an avidin–biotin–peroxidase complex for leptin, and then the slides were counterstained with hematoxylin.

The assessment of leptin receptor expression in cancer tissue was carried out by two separate pathologists. The expression of leptin was analyzed using light microscopy. The immunostaining for leptin was analyzed in 10 different fields. The mean percentage of tumor cells with positive staining was calculated. Positively colored cells were counted in 10 representative high-power fields and classified as follows: negative (−) with ≤ 10% positive cells, positive (+) with 11–49% positive cells (focal and moderate expression), and highly positive (++) with ≥50% positive cells (strong and diffuse expression). The counts were conducted in a set of 10 random fields at ×20 magnification.

### 2.2. Statistical Analysis

The Mann-Whitney test, Kruskal-Wallis test, Student’s *t*-test, chi-square test of independence, Spearman’s rank correlation coefficient, and multivariate variance analysis were used for statistical analysis [23]. The statistical package STATISTICA 13 by Statsoft was used for the calculations.

## 3. Results

### 3.1. Characteristics of Study Group

The study included 61 patients diagnosed with CRC; in the study group, there were 63.9% male patients and 36.1% female patients. The average age of all subjects was 70.5 ± 10.3 years, and the average BMI was 27.7. It was observed that 29.5% (18 people) of the study group were obese, 34.4% (21 people) were overweight, and 36.1% (22 people) had a normal body mass index (BMI). In the study group, more than half of the surveyed patients (57.4%: 35 people) smoked cigarettes; alcohol was consumed by just over a fifth of patients (21.3%: 13 people).

More than half of the people in the study group had hypertension, one in three had type II diabetes, about one in six people had cardiac arrhythmia and coronary heart disease, and one in ten men had benign prostatic hyperplasia. Approximately one in six people was found to have no comorbidity.

### 3.2. Leptin Receptor Expression

The study group was dominated by patients whose tumor size was over 3 cm. This size of the tumor was found in 49.2% of the subjects in the study group. All patients in the study group had positive leptin receptor expression in the tumor tissue. Almost 60% of patients had strong (++) leptin prescription expression. Leptin receptor expression was undetectable in the study samples of patients in the comparison group.

No leptin receptor expression was detected in the normal intestinal mucosa in all 20 tested scraps of normal intestinal mucosa. However, no significant correlation was found between leptin receptor expression and clinical–pathological parameters (Table 1).

Figure 1 shows the observed incidences of each leptin receptor expression level observed in both groups, along with 95% confidence intervals. In the 20-person comparison group, no case of moderate or strong leptin receptor expression was observed. It does not necessarily result in the absence of such cases in a larger population, as can be seen from the chart below.

With 95% confidence, it may be said that the percentage of such cases in the entire population of people with other diseases of the large intestine is no more than approximately 18%. Similarly, with 95% confidence, it can be concluded that in the group of patients with CRC, the occurrence of a low expression of the leptin receptor concerns a percentage of less than approximately 7% (*p* = 0.00).

For factors of a numerical (e.g., age and BMI) or ordinal (TNM stage) nature, Spearman’s rank correlation coefficient was used to analyze their effect on leptin levels. In the case of comparing the concentration of leptin with the levels of the grouping factor (e.g., expression of the leptin receptor), the analysis consisted of determining descriptive statistics in the compared groups and assessing the differences between the groups using the Mann-Whitney test (for two groups) or its generalization—the Kruskal-Wallis test (for more than two groups).

Among the cancerous tissues, all of the 61 (100%) samples were stained positive or highly positive via the immunohistochemical technique. In contrast, all of the 20 normal colorectal tissue were negatively stained. The difference in the leptin receptor expressions between the cancerous and normal colorectal mucosa were highly significant (*p* < 0.01, chi-squared). The results of this study suggest that leptin may be involved in the development of colorectal cancer.

### 3.3. Serum Leptin Concentrations

Table 2 provides information on the serum leptin concentrations; since the level of this hormone depends on sex, detailed descriptive statistics are stratified for sex.

The average leptin concentration among women is more than 2.5 times higher than that among men. The above summary shows that 42.6% of patients had above average leptin concentrations. For men, in the case of a higher expression of the leptin receptor, there is also a much higher concentration of leptin (the median leptin concentration value for a moderate expression of the leptin receptor is 3.40 ng/mL, and for a strong one, 6.23 ng/mL). This difference is statistically significant (*p* = 0.0032) (Figure 2).

Figure 3 shows the relationship between the age of patients and serum leptin concentration; the analysis was carried out taking into account the sex variable. A statistically significant correlation was found between age and leptin concentration among men: with age, leptin concentration decreases. However, this is not a very strong dependency (*r_s_* = −0.33).

Similarly, the results of the correlation analysis between BMI and the concentration of leptin are presented in Figure 4 and Table 3. Statistically significant and relatively strong (and for women, even very strong *r_s_* = −0.83) relationships between BMI and leptin concentration were obtained. The higher the BMI, the higher the concentration of leptin in the blood serum.

The analyses show that the concentration of leptin did not correlate statistically significantly with the degree of histological differentiation, lymph node metastases, the stage of TNM, tumor size, the occurrence of cancer in the family, or lifestyle. Type II diabetes is also not a statistically significant factor affecting leptin levels.

It was also examined whether the percentage of people who experienced above-average leptin concentrations depended on selected factors. Since the average leptin concentration takes into account the specificity of sex, the analysis can be carried out at the level of the whole community, including both women and men. To assess the significance of differences between groups in the participation of individuals with an oversized concentration of leptin, a chi-squared independence test was used.

Strong leptin receptor expression and the manifestation of overweight and obesity are factors influencing the occurrence of excessive leptin concentrations. Both of these relationships are statistically significant (test probability value *p* < 0.05) (Table 4). 

First, a comparison of leptin concentrations in the study and comparative groups was performed based on the Mann-Whitney test. The analysis was carried out separately for men and women. In this analysis, it was not possible to find statistically significant differences in the level of leptin concentration in the study and comparison groups (Table 5).

In the proposed approach, the difference between the groups was also statistically significant. Based on the value of descriptive statistics, it can be concluded that the concentration of leptin is significantly higher in the comparison group than among people with CRC (Figure 5).

The factor associated with leptin concentration is BMI. The study used a regression model with three variables: being a member of the study or comparative group, sex, and BMI.

The calculation was carried out for the concentration of leptin subjected to a logarithmic transformation (Table 6). The results of the analysis are presented below. It is evident that the concentration of leptin is higher for women and increases with BMI. On the other hand, the negative coefficient and high significance of the difference between the study and comparative groups confirm previous analyzes, which resulted in a lower concentration of leptin in the study group.

The regression model formula for the log leptin concentration is as follows:LOG (leptin concentrations) = −1.42 − 0.318·Group (study vs. comparative) + 1.075 Gender (Female vs. Male) + 0.112·BMI

However, after performing the inverse logarithmic transformation, the following form of this equation is obtained:Leptin concentration = 0.24 (Study Group) ^0.728^ (Female) ^2.930^·(BMI)1^.119^

The above results can be interpreted as follows: in women, the concentration of leptin is 2.93 times higher than that in men; an increase in BMI of 1 causes an elevation of 1.12 times in leptin concentration; and in the study group, the concentration of leptin is 0.728 (i.e., by 27.2%) less than in the comparison group. An illustration of the results of the above analyses can be found in the following figures, which show the relationship between BMI and leptin concentration (in logarithmic form) in the study and comparative groups—separately for women and men. The graphs below confirm the fact that there is a lower concentration of leptin in the study group, although they also allow us to see other interesting regularities, i.e., that the concentration of leptin is lower in the group of women studied for low BMI values, while in the case of a high BMI, such a difference does not occur.

However, for men, the situation is the opposite—for low BMI values, the difference in leptin concentration between the test and comparison groups is small, while the lower level of leptin concentration in the study group appears with increasing BMI values (Figure 6).

## 4. Discussion

Epidemiological data indicate that obesity is a risk factor for the development of CRC [5,10,24]. However, the possible molecular mechanisms responsible for this phenomenon are unclear. In this context, the correlation between the obesity hormone leptin and CRC has been studied in recent years. Studies using animal models and epidemiological data have shown controversial results, with a positive or negative correlation between serum leptin concentrations and CRC. In addition, some research has indicated that serum leptin levels are not linked with CRC [24,25]. Leptin is a peptide hormone with a molecular weight of 16 kD, which in adults is produced mainly in adipose tissue. At the same time, leptin, in much smaller amounts, is also secreted in numerous extrafatual tissues: the lungs, breast, gastric mucosa, brain, placenta, prostate, testicles, ovaries, and endometrium. In the study of Zaha et al. describe the potential role of leptin in inducing pro-inflammatory markers expression. C reactive protein is associated only with obesity, not with the metabolic syndrome. Therefore, assessment of adiponectin in the population could help identify patients with high risk of diabetes mellitus and cardiovascular disease [26,27,28,29,30,31]. Leptin is released cyclically, usually 2–3 h after a meal, and its serum concentration is directly correlated with the amount of body fat. It has been discovered that when there is an increase in the number and size of adipocytes, the leptin gene begins its production, which is then secreted into the bloodstream. Numerous reports from the literature show that leptin plays a crucial role in the progression and pathogenesis of CRC [32,33,34], while the research of Tutino et al. [34] has shown that high serum leptin levels are an independent risk factor for the development of CRC. Leptin receptor expression occurs in many cancer cells, including colorectal cancer cells. Literature reports in recent years show that leptin receptor expression is positive in approximately 77 to 95.5% of patients with CRC [19,35,36,37,38].

Obesity increases the risk of cancer formation through molecular mechanisms resulting from excessive amounts of adipose tissue and through the coexistence of hyperinsulinemia and hyperlipidemia directly related to lifestyle. Adipose tissue is composed of subcutaneous fat secreting large amounts of leptin and visceral fat, which are more hormonally active and secrete biologically active compounds, i.e., adiponectin, IL-6, and resistin [39,40], TNF-α (tumor necrosis factor), visfatin, or PAI-11 (plasma activator inhibitor). Under the physiological conditions of normal body weight, the appropriate proportions between individual substances are maintained, which is beneficial for human health and enables the proper functioning of the body. In obesity, there is an imbalance between them, which may result, among other complications, in the development of a chronic inflammatory process and insulin resistance [39,40,41]. Many pro-inflammatory cytokines increase the concentration of TNFα and IL-6, which are the key cytokines in cancer progression. Tumor necrosis factor activates the nuclear factor (NF-kB nuclear factor) by binding to the TNF receptor, blocking apoptosis, and increasing the proliferation of neoplastic cells. Interleukin 6 sends signals to the cell nucleus via a signal transducer to Transcription Activator 3 (STAT3), an oncoprotein activated in many cancerous tumors. The excess of adipose tissue is one of the factors determining carcinogenesis in obesity [39,40,41].

The second factor is the lifestyle of overweight and obese people—a lack of physical activity and excess energy supplied from food in relation to the demand and quality of meals consumed. A study [42] of nearly 3500 adults showed that obese people lead a less active lifestyle compared with people with a normal body weight. Physical activity is an indispensable element of cancer prevention as it influences the immune system by alleviating inflammation, lowering the concentration of sex hormones, reducing body weight, and improving intestinal peristalsis [42]. In the study of S Vuletic et al. [26], the intracytoplasmic and intramembrane expression of the leptin receptor was verified in a significant number of cases (77.3%), with pronounced leptin receptor expression in approximately one-third of the cases (33.3%). The study [26] showed that leptin receptor expression was significantly associated with lymph node metastases and distant metastases. There was no significant association [26] of leptin receptor expression with the patient’s demographic characteristics, which was consistent with the results of Wang et al. [34], who studied leptin receptor expression in colorectal cancer with regard to demographic parameters. The researchers Koda et al. [43] showed a statistically significant positive correlation of leptin receptor expression with female sex and an age of over 60 years. The study of Al-Shibli et al. [44] further highlights the possible role of leptin receptor expression in CRC, as well as the prospect of using leptin receptor expression as a possible therapeutic target. Research by Al-Shibli et al. [44] is consistent with the research of Koda et al. [45]. To date, there is no other report that has found such a frequent occurrence of leptin receptor expression in CRC or any other cancer. The closest numerical incidence of leptin receptor expression was reported by Al-Maghrabi et al. [46], where positive leptin receptor expression was observed in 93.5% of cases of CRC in the Western Province of Saudi Arabia. Koda et al. [45] reported that leptin cells are overexpressed in CRC relative to normal colon mucosa in their study. Leptin is associated with carcinogenesis and the progression of various cancers. However, changes in serum leptin levels in patients with CRC and their relationship to the treatment response in these patients have rarely been studied [47,48]. In the study of Wang et al. [49] on CRC, a correlation was found between the serum leptin concentration and focal leptin receptor expression in tumor tissue. The serum leptin concentrations of patients with CRC were significantly higher compared with those of the control group (22.67 ± 12.56 vs. 12.68 ± 7.8 ng/mL; *p* < 0.05, respectively). In addition, leptin levels after surgery decreased compared with preoperative levels (18.67 ± 8.54 vs. 22.67 ± 12.56 ng/mL; *p* < 0.05, respectively). In summary, leptin levels were elevated in overweight and colorectal cancer patients. Leptin concentration decreased after colectomy, indicating that leptin may be associated with colon carcinogenesis. Research by Wang et al. [49] argues that serum leptin levels can be used for early diagnosis and the monitoring of colorectal cancer treatment response. Our studies did not show a positive correlation between leptin concentrations in patients with CRC and those in the comparison group. There were also no statistically significant differences between the leptin concentration and leptin receptor expression. The purpose of the study of Salagean et al. [50] was to investigate the relationship between several levels of adipocytokines in the blood and the clinical–pathological characteristics of colorectal cancer patients undergoing surgery. Resistin levels were significantly higher in patients with colon cancer, while serum leptin levels were significantly lower compared with the controls. Leptin levels dropped gradually as the tumor progressed. In conclusion, the results of this study suggest that adipokines, in particular, resistin and leptin, may be involved in the development and progression of CRC. The relationship between serum leptin levels and the risk of CRC remains controversial. In the study of Wang et al. [49], significantly lower serum leptin concentrations were found in patients with CRC in contrast to the control group. In addition, a significant correlation was observed between serum leptin concentration and cancer staging based on TNM classification (*p* = 0.021). 

### The Limitation of the Study

Obesity increases the risk of many types of cancer, including colorectal cancer. Although the molecular mechanisms underlying this compound are not known. The results of this study also do not demonstrate the molecular mechanism. The issue is that many things are altered by obesity. But it is very hard to know if the changes that are being observed are due to changes in leptin and leptin receptors, or something else entirely. For instance, other factors produced by adipose tissue are also likely to be increased and could also be driving the changes that are observed independently of leptin. Thus, in vitro experiments on colorectal cancer cells would be necessary to observe the effect of leptin on cancer development and progression.

Above studies is the small size of the study group and the comparison group. Perhaps a statistical analysis of a larger group of colorectal cancer patients would lead to more precise conclusions.

## 5. Conclusions

In conclusion, the results suggest that leptin may be involved in the development and progression of colorectal cancer. In order to better clarify the role of leptin in the development and progression of the disease, more research is needed.

## Figures and Tables

**Figure 1 ijerph-20-04951-f001:**
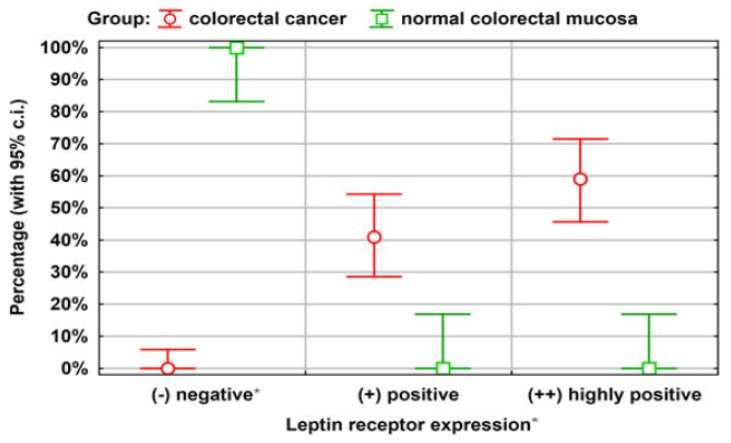
Leptin receptor expression among patients in the study group (*n* = 61) and comparison group (*n* = 20). Statistical analysis was performed using the Kruskal-Wallis test and the Mann-Whitney test; * *p* = 0.001, CI (confidence interval) was calculated by Student’s *t*-test.

**Figure 2 ijerph-20-04951-f002:**
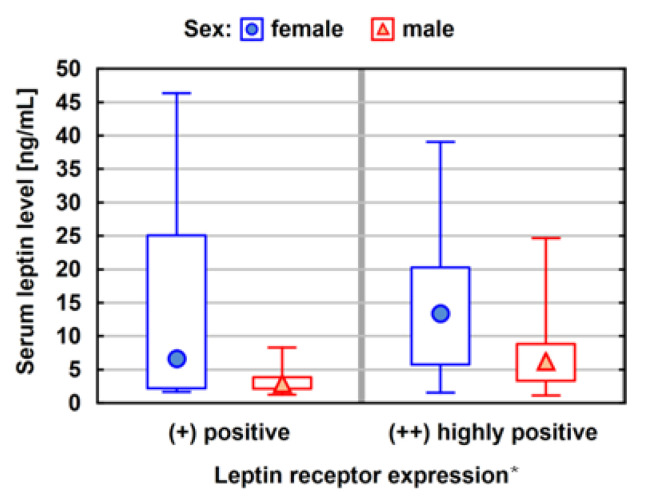
Median values with lower and upper quartiles and minimum and a maximum leptin concentrations for both sexes relative to prescription leptin expression (N = 61). Statistical analysis was performed using the Kruskal-Wallis test and the Mann-Whitney test; *p*-value < 0.05, * *p* = 0.0032.

**Figure 3 ijerph-20-04951-f003:**
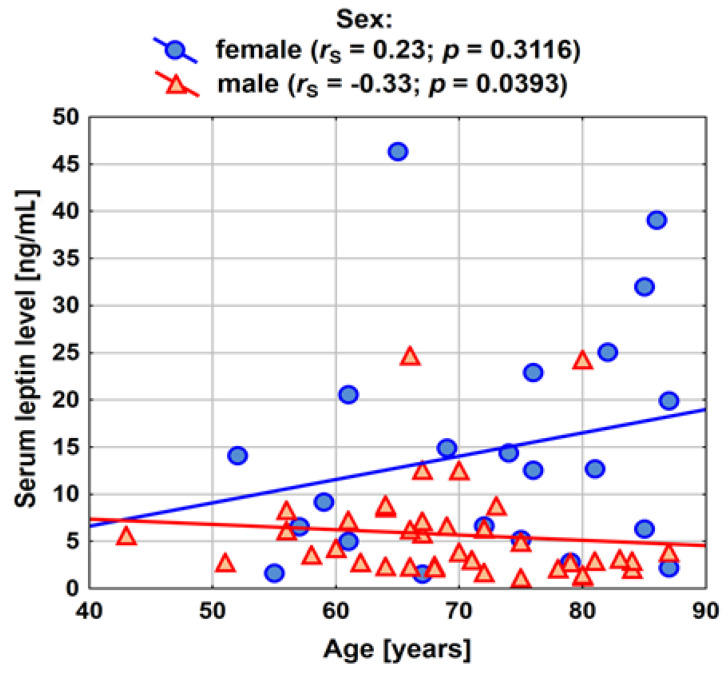
Spearman’s rank correlation between age and leptin concentration among patients in the study group (male: *n* = 39; female: *n* = 22).

**Figure 4 ijerph-20-04951-f004:**
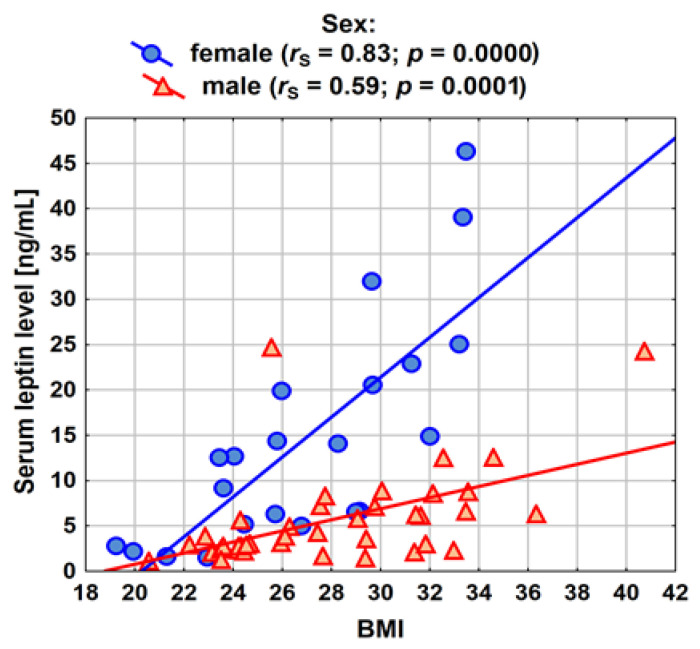
Spearman’ rank correlation between BMI and serum leptin concentration among patients in the study group (male: *n* = 39; female: *n* = 22).

**Figure 5 ijerph-20-04951-f005:**
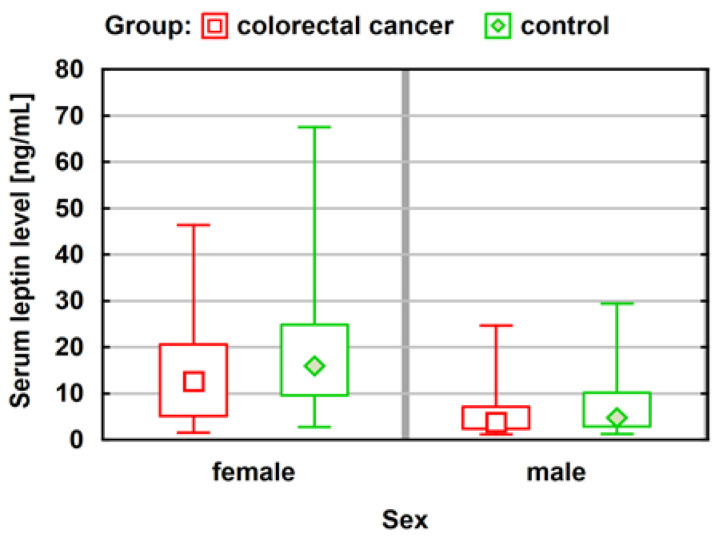
Comparison of leptin concentration in the study and comparison groups. Study group (*n* = 61); control group (*n* = 60). Statistical analysis was performed using the Kruskal-Wallis test and the Mann-Whitney test; *p*-value < 0.05.

**Figure 6 ijerph-20-04951-f006:**
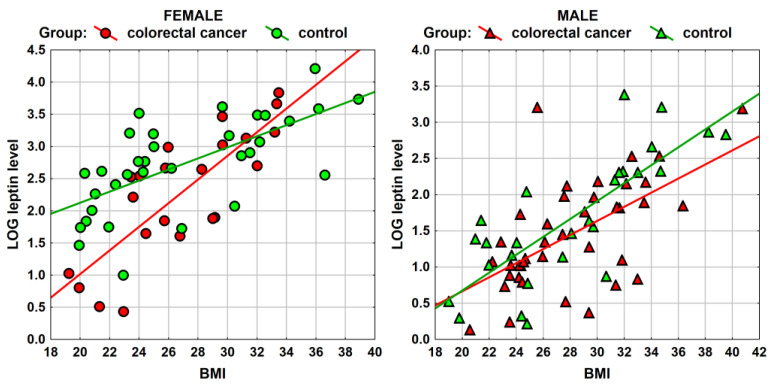
Relationship between BMI and leptin concentration in logarithmic form in the study and comparative group separately for females and males. Study group: female (*n* = 22); male (*n* = 39). Control group: female (*n* = 33); male (*n* = 27).

**Table 1 ijerph-20-04951-t001:** Influence of clinical-pathological factors on leptin receptor expression.

Clinical–Pathological Parameters		Leptin Receptor Expression		*p*-Value
		Moderate Expression (+)	Strong Expression (++)	
age	<65 yrs.	6 (35.3%)	11 (64.7%)	0.69
	65−74 yrs.	8 (38.1%)	13 (61.9%)	
	≥75 yrs.	11 (47.8%)	12 (52.2%)	
sex	female	6 (27.3%)	16 (72.7%)	0.10
	male	19 (48.7%)	20 (51.3%)	
Histological type	glandular cancer	9 (31)%	20 (69)%	0.78
	tubular type glandular carcinoma	7 (36.8)%	12 (63.2)%	
	mucous type glandular carcinoma	5 (62.5)%	3 (37.5)%	
	spindle cell type glandular cancer	4 (80)%	1 (20)%	
BMI	norm	10 (45.5%)	12 (54.5%)	0.72
	overweight	9 (42.9%)	12 (57.1%)	
	obesity	6 (33.3%)	12 (66.7%)	
Lymph node metastases	yes	10 (50.0%)	10 (50.0%)	0.32
	no	15 (36.6%)	26 (63.4%)	
Tumor size	<1 cm	3 (42.9%)	4 (57.1%)	0.79
	1–3 cm	11 (45.8%)	13 (54.2%)	
	>3 cm	11 (36.7%)	19 (63.3%)	
TNM level of advancement	I	5 (26.3%)	14 (73.7%)	0.24
	IIA	7 (41.2%)	10 (58.8%)	
	IIIA-IIIB	7 (43.8%)	9 (5.3%)	
	IVA	6 (66.7%)	3 (33.3%)	

Statistical analysis was performed using the Kruskal-Wallis test and the chi-squared independence test; *p*-value < 0.05.

**Table 2 ijerph-20-04951-t002:** The concentration of leptin in the blood serum among patients in the study group.

Sex	Serum Leptin Concentration [ng/mL]						
	$\bar{x}$	Me	*S*	*c* _25_	*c* _75_	min	max
Female	14.63	12.63	12.34	5.19	20.60	1.55	46.36
Male	5.71	3.85	5.29	2.42	7.12	1.14	24.68

Study group (*n* = 61) (male: *n* = 39; female: *n* = 22).

**Table 3 ijerph-20-04951-t003:** Effect of overweight and obesity on leptin concentration.

Serum Leptin Concentration	BMI (*p* = 0.00)			Total
	Normal *n* (%)	Overweight *n* (%)	Obesity *n* (%)	
Normal	21 (95.5%)	11 (52.4%)	3 (16.7%)	35
Above average	1 (4.5%)	10 (47.6%)	15 (83.3%)	26
Total	22	21	18	61

Statistical analysis was performed using the chi-squared independence test; *p*-value < 0.05.

**Table 4 ijerph-20-04951-t004:** Effect of leptin receptor expression on leptin concentration.

Leptin Concentration	Leptin Receptor Expression (*p* = 0.00)		Total *n*
	Moderate Expression (+)	Strong Expression (++)	
Normal	20 (80.0%)	15 (41.7%)	35
Above average	5 (20.0%)	21 (58.3%)	26
Total	25	36	61

Study group (*n* = 61). Statistical analysis was performed using the chi-squared independence test; *p*-value < 0.05.

**Table 5 ijerph-20-04951-t005:** Comparison of leptin concentration in the study and comparison groups.

Group					Serum Leptin Concentration [ng/mL]												Total
					Sex												
					Female (*p* = 0.13)				Male (*p* = 0.25)								N
	*n*	$\bar{x}$	Me	*c* _25_	*s*	*c* _75_	min	max	*n*	$\bar{x}$	Me	*s*	*c* _25_	*c* _75_	min	max	
Study group	22	14.63	12.63	5.19	12.34	20.60	1.55	46.36	39	5.71	3.85	5.29	2.42	7.12	1.14	24.68	61
Comparison group	33	19.44	15.90	5.85	13.80	21.40	1.85	47.60	27	7.80	4.75	7.28	2.90	8.20	1.80	26.10	60

Study group (*n* = 61); comparison group (*n* = 60). Statistical analysis was performed using the Mann-Whitney test; *p*-value < 0.05.

**Table 6 ijerph-20-04951-t006:** LOG leptin concentration in the study (N = 61) and comparison groups (*n* = 60).

Independent Factors	LOG Leptin Concentrations *R*^2^ = 62.9% *F* = 66.1 *p* = 0.0000		
	*B* (95% c.i.)	*p*	*β*
Group (studied vs. comparative)	−0.318 (−0.538–0.098)	0.00	−0.16
Sex (female vs. male)	1.075 (0.852–1.297)	0.00	0.55
BMI	0.112 (0.091–0.134)	0.00	0.58

## Data Availability

The raw data supporting the conclusions of this article will be made available by the authors, without undue reservation. Informed consent was obtained from all subjects involved in the study.

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
