# Peer review of "The Effect of Serum Leptin Concentration and Leptin Receptor Expression on Colorectal Cancer"

_ijerph, 2023, doi:10.3390/ijerph20064951_

Round 1
Reviewer 1 Report
This is a resubmission in which the authors addressed some but not all of the concerns of the reviewers.
Of note is the lack of photomicrographs showing actual leptin receptor expression to varying degrees in intestinal mucosa. These data are the basis for: Table 1, Figure 1, Figure 2 and Table 4. The revised manuscript does not contain these photomicrographs.
The authors did include statistical analyses for the tables and figures. However, further editing is required. Specifically, which treatment groups are statistically different from one another needs to be indicated in the table or figure itself. For example, in Figure 1 the X axis title “Leptin receptor expression*” contains an asterisk however, it is not known which groups are significantly different from one another on the figure. Are the mean values for the colorectal cancer vs normal colorectal samples with negative leptin expression significantly different? What about colorectal cancer vs normal colorectal samples for “+” positive or “++” positive leptin expression? It’s insufficient to indicate that significant differences exist in an entire graph or data set when the reader wants to know if statistically significant differences exist between specific treatment groups. The statistical tests used (Kruskal-Wallis and Mann-Whitney) would provide this type of information. In other words, if statistically significant differences exist between colorectal cancer vs normal colorectal samples with negative leptin expression on Figure 1 then indicate this with an asterisk above or below the error bars.
The stating that a p value is equal to 0.00 on lines 170 and 172 is not accurate. Change to “p < 0.001” if that is accurate and remove “p < 0.05”. If the p value is less than 0.001 it is unnecessary to also state that the p value is less than 0.05.
Minor text editing is needed throughout the manuscript. For example line 105 “wo” should be “who”.
Author Response
Reviwer 1
This is a resubmission in which the authors addressed some but not all of the concerns of the reviewers
Replay: The manuscript is submitted to IJERPH for the first time.
Of note is the lack of photomicrographs showing actual leptin receptor expression to varying degrees in intestinal mucosa. These data are the basis for: Table 1, Figure 1, Figure 2 and Table 4. The revised manuscript does not contain these photomicrographs.
Replay: We don't have photomicrographs, they've been in the lab and are hard to get now. The authors confirm that the research was conducted reliably.
The authors did include statistical analyses for the tables and figures. However, further editing is required. Specifically, which treatment groups are statistically different from one another needs to be indicated in the table or figure itself. For example, in Figure 1 the X axis title “Leptin receptor expression*” contains an asterisk however, it is not known which groups are significantly different from one another on the figure. Are the mean values for the colorectal cancer vs normal colorectal samples with negative leptin expression significantly different? What about colorectal cancer vs normal colorectal samples for “+” positive or “++” positive leptin expression? It’s insufficient to indicate that significant differences exist in an entire graph or data set when the reader wants to know if statistically significant differences exist between specific treatment groups. The statistical tests used (Kruskal-Wallis and Mann-Whitney) would provide this type of information. In other words, if statistically significant differences exist between colorectal cancer vs normal colorectal samples with negative leptin expression on Figure 1 then indicate this with an asterisk above or below the error bars.
Replay: corrected
The stating that a p value is equal to 0.00 on lines 170 and 172 is not accurate. Change to “p < 0.001” if that is accurate and remove “p < 0.05”. If the p value is less than 0.001 it is unnecessary to also state that the p value is less than 0.05.
Replay: corrected
Minor text editing is needed throughout the manuscript. For example line 105 “wo” should be “who”.
Replay: corrected
Reviewer 2 Report
L61, 67, 72. 3 consecutive references should be written [10-12, [5, 13-15], [15-17] etc. Please revise the entire manuscript in this regard.
L131. The computer soft used must be referenced. I suggest checking https://libguides.library.kent.edu/statconsulting/software-help and proceed consequently.
The Discussion chapter can be improved. Please describe previous studies that demonstrated that weight loss can be an important method in reducing the incidence of colorectal cancer. Please make a summary table including at least 8-10 studies that referred to a large number of individuals. Also describe the potential role of leptin in inducing pro-inflammatory markers expression – I suggest checking and referring to PMID: 32509004
Please describe the possibility of using leptin in current clinical practice for the grading of colorectal cancer severity. Please describe wheter leptin expression correlates with expression of other important colorectal cancer biomarkers such as p53 expression ATP-ase, LDH, how they interact and what molecular patterns can be described.
A last section of Conclusions would be relevant, moving here paragraph L385-387 and improving it.
Author Response
Reviwer 2
- L61, 67, 72. 3 consecutive references should be written [10-12, [5, 13-15], [15-17] etc. Please revise the entire manuscript in this regard.L131.L61, 67, 72. Należy wpisać 3 kolejne pozycje piśmiennictwa [10-12, [5, 13-15], [15-17] itd. Proszę o przejrzenie całego rękopisu pod tym kątem.
Replay: corrected
- The computer soft used must be referenced. I suggest checking https://libguides.library.kent.edu/statconsulting/software-help and proceed consequently. L131.
Replay: We added a reference: Lomax, RG.; Hahs-Vaughn, DL. Statistical concepts : a second course. Routledge,New York, 2012.
- The Discussion chapter can be improved. Please describe previous studies that demonstrated that weight loss can be an important method in reducing the incidence of colorectal cancer. Please make a summary table including at least 8-10 studies that referred to a large number of individuals. Also describe the potential role of leptin in inducing pro-inflammatory markers expression – I suggest checking and referring to PMID: 32509004
Replay: It was add: In the study of Zaha et al. describe the potential role of leptin in inducing pro-inflammatory markers expression. C reactive protein is associated only with obesity, not with the metabolic syndrome. Therefore, assessment of adiponectin in population could help identify patients with high risk of diabetes mellitus and cardiovascular disease.
Zaha DC, Vesa C, Uivarosan D, Bratu O, Fratila O, Tit DM, Pantis C, Diaconu CC and Bungau S: Influence of inflammation and adipocyte biochemical markers on the components of metabolic syndrome. Exp Ther Med 20: 121-128, 2020
With all due respect to the person of the reviewer, adding a table with selected studies now disturbs our methodological approach to research. Such presentation of the results requires a systematic review of the literature, while maintaining the criteria for inclusion and exclusion of given studies. This may be an idea for further research.
- Please describe the possibility of using leptin in current clinical practice for the grading of colorectal cancer severity. Please describe wheter leptin expression correlates with expression of other important colorectal cancer biomarkers such as p53 expression ATP-ase, LDH, how they interact and what molecular patterns can be described.
Replay: In the discussion, we refer to the role of leptin in CRC: Epidemiological data indicate that obesity is a risk factor for the development of CRC [5,10,24]. However, the possible molecular mechanisms responsible for this phenomenon are unclear. In this context, the correlation between the obesity hormone leptin and CRC has been studied in recent years. Studies using animal models and epidemiological data have shown controversial results, with a positive or negative correlation between serum leptin concentrations and CRC. In addition, some research has indicated that serum leptin levels are not linked with CRC
- A last section of Conclusions would be relevant, moving here paragraph L385-387 and improving it.
Replay: corrected